DBFU-Net: Double branch fusion U-Net with hard example weighting train strategy to segment retinal vessel

Huang Jianping 1
Lin Zefang 1
Chen Yingyin 1
Zhang Xiao 1
Zhao Wei 1
Zhang Jie 2
Li Yong 1
He Xu 1
Zhan Meixiao 1
Lu Ligong 1
Jiang Xiaofei 3
Peng Yongjun pengyongjun_zy@163.com 2
1 Zhuhai People’s Hospital, Zhuhai Hospital Affiliated with Jinan University, Jinan University, Zhuhai Interventional Medical Center, Zhuhai Precision Medical Center , Zhuhai , China
2 Zhuhai People’s Hospital, Zhuhai Hospital Affiliated with Jinan University, Jinan University, Department of Nuclear Medicine , Zhuhai , China
3 Zhuhai People’s Hospital, Zhuhai Hospital Affiliated with Jinan University, Jinan University, Department of cardiology , Zhuhai , China
Tsai Yi-Hsuan
Electronic publication date: 2022 Feb 18
Publication date: 2022
Volume: 8
Electronic Location ID: e871
Received 2021 Jan 25; Accepted 2022 Jan 10
Copyright: ©2022 Huang et al.
Copyright year: 2022
Copyright holder: Huang et al.
License: This is an open access article distributed under the terms of the Creative Commons Attribution License, which permits unrestricted use, distribution, reproduction and adaptation in any medium and for any purpose provided that it is properly attributed. For attribution, the original author(s), title, publication source (PeerJ Computer Science) and either DOI or URL of the article must be cited.
License URL: https://creativecommons.org/licenses/by/4.0/

Keywords: Vessel segmentation, Deep learning, Fundus image, Hard examples, Random channel attention mechanism

Funding: The National Key Research and Development Program of China 2017YFA0205200 The National Natural Science Foundation of China 81901857 The Natural Science Foundation of Guangdong Province, China 2018A030313074 This study is supported by the National Key Research and Development Program of China (Grant No. 2017YFA0205200), the National Natural Science Foundation of China (Grant No. 81901857), and the Natural Science Foundation of Guangdong Province, China (No. 2018A030313074). The funders had no role in study design, data collection and analysis, decision to publish, or preparation of the manuscript.

==============================
Background

Many fundus imaging modalities measure ocular changes. Automatic retinal vessel segmentation (RVS) is a significant fundus image-based method for the diagnosis of ophthalmologic diseases. However, precise vessel segmentation is a challenging task when detecting micro-changes in fundus images, e.g., tiny vessels, vessel edges, vessel lesions and optic disc edges.

Methods

In this paper, we will introduce a novel double branch fusion U-Net model that allows one of the branches to be trained by a weighting scheme that emphasizes harder examples to improve the overall segmentation performance. A new mask, we call a hard example mask, is needed for those examples that include a weighting strategy that is different from other methods. The method we propose extracts the hard example mask by morphology, meaning that the hard example mask does not need any rough segmentation model. To alleviate overfitting, we propose a random channel attention mechanism that is better than the drop-out method or the L2-regularization method in RVS.

Results

We have verified the proposed approach on the DRIVE, STARE and CHASE datasets to quantify the performance metrics. Compared to other existing approaches, using those dataset platforms, the proposed approach has competitive performance metrics. (DRIVE: F1-Score = 0.8289, G-Mean = 0.8995, AUC = 0.9811; STARE: F1-Score = 0.8501, G-Mean = 0.9198, AUC = 0.9892; CHASE: F1-Score = 0.8375, G-Mean = 0.9138, AUC = 0.9879).

Discussion

The segmentation results showed that DBFU-Net with RCA achieves competitive performance in three RVS datasets. Additionally, the proposed morphological-based extraction method for hard examples can reduce the computational cost. Finally, the random channel attention mechanism proposed in this paper has proven to be more effective than other regularization methods in the RVS task.

Introduction

Diabetic retinopathy (DR) refers to progressive retinal damage that occurs in people with diabetes. This disease may cause vision loss, has no symptoms in the early stages, and usually develops rapidly (Yin et al., 2015). The narrowing of small blood vessels in the retina is a specific indicator of the disease, thus the ophthalmologists can make a diagnosis by analyzing the retinopathy (Staal et al., 2004). However, due to the high prevalence of diabetes and the lack of human experts, screening procedures are expensive and time-consuming for clinics. Thus, reliable automatic analysis methods of retinal images will greatly reduce the workload of ophthalmologists and contribute to a more effective screening procedure (Azzopardi, Vento & Petkov, 2015; Wang et al., 2015). Therefore, a computer-aided automated retinal vessel segmentation (RVS) is highly desirable in many cases (Ricci, 2007).

Automated RVS is a well-regarded method in ophthalmologic image analysis. Automatic computer-aided medical image analysis has been introduced to improve the performance and efficiency of RVS in recent years, thanks to advances in image processing and artificial intelligence. We divide these methods into two categories: learning-based and non-learning-based methods. Machine learning-based methods can further be categorized as supervised and unsupervised methods. The algorithm we propose in this paper is a supervised deep learning method.

Related work

The retinal vessel extraction problem is comparable to the segmentation of foreground and background in fundus image. The related research can be traced back to the late 1980’s (Chaudhuri et al., 1989). In recent years, machine learning methods have become more popular and successful in natural image processing. An increasing number of medical image research projects have focused on learning-based algorithms.

Non-learning-based methods

Non-learning-based segmentation methods are often limited to an accurate artificial description capability, while learning-based methods are limited to training data (Li et al., 2018). For example, Sheng et al. (2019) proposed a robust and effective approach that qualitatively improves the detection of low-contrast and narrow vessels. Rather than using the pixel grid, they used a super-pixel as the elementary unit of the vessel segmentation scheme. Khan et al. (2019) presented a couple of contrast-sensitive measures to boost the sensitivity of existing RVS algorithms. They applied a scale-normalized detector that detects vessels regardless of size. A flood-filled reconstruction strategy was adopted to get a binary output. Sazak, Nelson & Obara (2019) introduced a new approach based on mathematical morphology for vessel enhancement, which combines different structuring elements to detect the innate features of vessel-like structures. The non-learning-based methods can avoid complex training processes, but their segmentation performance is not as good as learning-based algorithms.

Unsupervised methods

An unsupervised learning algorithm can be executed without requiring manual annotations. Currently, most RVS tasks (based on unsupervised methods) in the literature are performed using Hessian matrices, morphological operations, or similar methods. Neto et al. used statistics of spatial dependency and probability to coarsely approximate a vessel map with an adaptive local thresholding scheme. The coarse segmentation is then refined through curvature analysis and morphological reconstruction to reduce pixel mislabeling and to better estimate the retinal vessel tree (Câmara Neto et al., 2017). Zhang et al. (2020) proposed an accelerated matrix decomposition mechanism, which could be used to boost not only the original Hessian-based multi-scale approach, but also singular value decomposition-based algorithms. Yin et al. (2015) used an orientation-aware detector (OAD) to accurately extract retinal vessels. The detector has a linearly elongated structure and was designed according to the locally oriented intrinsic property of vessels Yin et al. (2015). Researchers used the OAD to extract the shape of a vessel with no assumptions on the parametric orientations of the vessel shape. Thus, the various orientations of vessels can be efficiently modeled by an energy distribution of a Fourier transformation. The major advantage of using unsupervised algorithms is that they do not require manual labeling for training, however their performance is not as good as most supervised algorithms.

Supervised methods

Generally, the performance of supervised segmentation methods is better than that of unsupervised methods, generally because these methods were all based on already classified data for segmentation (Akbar et al., 2019). The supervised learning-based approaches can further be classified into two groups: shallow learning-based approaches and artificial neural network-based approaches. Currently, shallow learning-based segmentation methods utilize handcrafted features for prediction. Palanivel, Natarajan & Gopalakrishnan (2020) proposed a novel retinal vasculature segmentation method based on multifractal characterization of the vessels to minimize noise and enhance the vessels during segmentation. The Holder exponents are computed from Gabor wavelet responses, which is an effective way to segment vessels and a novel feature of the method. However, the local regularity of the vessel structures extracted, based on Holder exponents, can easily miss small vessels features.

In fundus imaging, artificial neural networks were first used for classification tasks (Akita & Kuga, 1982). After the introduction of FCNs, a growing number of researchers turned to deep convolutional neural networks for segmentation tasks. Since then, several attempts have been made by introducing deep convolution neural network framework algorithms to segment retinal vessels. Yang et al. (2020) proposed a multi-scale feature fusion RVS model based on U-Net, called MSFFU-Net, that introduces an inception structure into the multi-scale feature extraction encoder part of the process. Additionally, a max-pooling index was applied during the upsampling process in the feature fusion decoder of the improved network. Leopold et al. (2017) had compiled various key performance indicators (KPIs) and state-of-the-art methods that were applied to the RVS task; this framed computational efficiency–performance trade-offs under varying degrees of information loss using common datasets and introduced PixelBNN. Thus, highly efficient deep learning methods for automating the segmentation of fundus morphologies was discovered. A retinal image segmentation method is also proposed by Li et al., called the MAU-Net (Li et al., 2020), that takes advantage of both modulated deformable convolutions and dual attention modules to realize vessel segmentation based on the U-net structure. Kromm & Rohr (2020) developed a novel deep learning method for vessel segmentation and centerline extraction of retinal blood vessels based on the Capsule network in combination with an Inception architecture. Ribeiro, Lopes & Silva (2019) explored the implementation of two ensemble techniques for RVS, Stochastic Weight Averaging and Snapshot Ensembles. Adarsh et al. (2020) implemented an auto encoder deep learning network model based on residual paths and a U-Net that effectively segmented retinal blood vessels. Guo et al. presented a multi-scale supervised deep learning network with short connections (BTS-DSN) (Guo et al., 2019) for vessel segmentation. Researchers used short connections to transfer semantic information between side-output layers. Bottom-top short connections pass from low-level semantic information to high-level information to refine the results to high-level side-outputs. The top-bottom short connection transmits structural information to the low-level to reduce the noise of low-level side-outputs. Yan, Yang & Cheng (2019) explored the segments of both thick and thin vessels separately by proposing a three-stage deep learning model. The vessel segmentation task is divided into three stages: thick vessel segmentation, thin vessel segmentation, and vessel fusion. Zhao et al. proposed a new approach as a step of post-processing (Zhao, Li & Cheng, 2020) to improve the existing method by formulating the segmentation as a matting problem. A trimap is obtained via a bi-level thresholding of the score map using existing methods, which is instrumental in focusing the attention to the pixels of these unknown areas. Among these ANN methods, (Yang et al., 2020; Kromm & Rohr, 2020; Adarsh et al., 2020; Guo et al., 2019; Yan, Yang & Cheng, 2019) have researched on multi-scale features, (Li et al., 2020) has researched attention mechanisms, (Ribeiro, Lopes & Silva, 2019) has researched ensemble strategy methods, Leopold et al. (2017) has researched the dependency between pixels, and (Zhao, Li & Cheng, 2020) has researched post-processing methods. These studies can improve the accuracy of segmentation models. But they did not focus on many difficult samples in the training process, so the overall segmentation performance (F1 Score) of these methods is not good enough.

Current supervised algorithms have produced some excellent results in RVS. The segmentation performance on optic disc, thin vascular, and lesion areas, however, remain unsatisfactory. The output probability map of models in optic disc, thin vascular, and lesion areas are close to 0.5, and thus we call these examples hard examples. The mask of hard examples can guide the training process of the models. Current methods to extract hard example masks, however, need rough segmentation results to set a probability range for increasing the computational complexity of the algorithm. In addition, the data volume of RVS datasets, which are scarce, will lead to problems of model overfitting, limiting the use of deep learning algorithms.

Contributions

To overcome problems described at 1.1.3, we directly extract the hard example mask from the ground truth via morphology. Then, we matched the hard example masks to design the double branch fusion U-Net (DBFU-Net), where one branch was trained by cross entropy and the other branch was trained by the improved cross entropy that applies weights to the hard example. In addition, we propose a random channel attention mechanism to prevent overfitting. The main contributions of this paper are listed as follows:

To overcome overfitting, we propose a novel regularization method, called Random Channel Attention Mechanism (RCA), that applies random weights to hidden layers channel-wise. The performance of the proposed regularization method is better than dropout and L2 regularization.

To extract the hard example of RVS, we propose a hard example extraction method based on image morphology.

We propose a DBFU-Net that fuses with two decoder branches, such that one of the branches pays more attention to the hard example to improve the segmentation performance.

As an overview, the details of the proposed method are introduced in Section 2. Section 3 describes the experimental process and discusses the segmentation results. The conclusion of the paper is provided in Section 4.

Methods

RVS is challenging when applying deep learning of models of the optic disc, thin vascular, and lesion region; this is largely because the pixels of these areas are not distinct from each other. Furthermore, the model is subject to overfitting during the training process due to data scarcity.

To improve segmentation performance in a hard example and alleviate overfitting, we propose the DBFU-Net trained by RCA. The DBFU-Net training process requires a hard example mask. We propose a hard example mask extraction method based on morphology to reduce computational cost. In this section, we define the RVS hard example first. Then, we describe the hard example mask extraction method based on morphology and the weighting loss of hard examples. After that, we introduce the RCA regularization approach and the structural details of DBFU-Net. Finally, we describe the implementation of our method.

Hard example extraction base on morphology

Generally, the loss of segmentation result is computed via cross entropy in the end-to-end model train process. Each pixel is treated with equal importance, however, the hard segmentation region cannot be more important. To ensure the model is more focused on the hard segmentation area, we could extract hard example masks and weight the loss function.

The output of the RVS model is usually a probability distribution map when the output value of the pixel probability is close to 1; in this case, the model considers the pixel to be a blood vessel. If the pixel probability value of the output is close to 0, the model considers the pixel to be part of the background. However, if the pixel probability value of the output is close to 0.5, it is difficult for the model to determine whether the pixel is a blood vessel or part of the background. Hence, we call all pixels that are difficult to distinguish via modeling hard examples, described by a probability values range.

Based on the definition of hard examples of RVS, Zhao et al. extracted the hard example mask by setting a threshold range (Zhao, Li & Cheng, 2020), shown in Fig. 1. However, hard example extractions need a probability map generated by the model. Hard example masks can guide the model to focus more on hard example areas in the training process; this is the reason why we would like to extract the hard example mask. Thus, a model is needed to extract these hard example masks extracted by a probability range to generate a rough segmentation result, leading to an increase in calculations and to a higher computational cost. To reduce the computational complexity of the hard example mask extraction, we propose a novel method that extracts the hard example mask of the RVS based on morphology, see Fig. 2. The details of the hard example mask are shown in Fig. 3.

Figures 2 and 3 demonstrates that, compared to the result of using the threshold-based method and the morphology operation-based method, the hard example mask extracted by both methods can obtain similar results; however the proposed method is simpler and more efficient than a probability threshold-based method. This is because a morphology operation-based method doesn’t need a probability map from a rough segmentation model. The hard example mask can be described by Eq. (1). (1) Mask=1,pixelisahardexample0,pixelisaeasyexample.

To train the model with more of a focus on the hard example, we set hard example loss weights, which means that we add a weighting value to the overall weighting of the cross-entropy loss function.

Figure 1 Hard example mask extraction via probability range.

Figure 2 Hard example masks extracted by morphology.

(A) Original label; (B) positive examples extracted by (A) after eroding; (C) hard examples of positive examples obtained by (A–B) that contain the edges of large vessels and thin vessels; (D) extracted via the dilated (C) and can further obtain hard examples among the negative samples, e.g., the edges of large vessels and the area near the thin vessels. However, the operation brought redundant non-hard positive examples; (E) is final hard example mask, that extract by (D–B), can remove the redundant non-hard positive examples.

Figure 3 The details of hard example masks are extracted by morphology-based method.

The white part is the label of the fundus image, and the red mask is a hard example mask extracted by the proposed method.

(2) Wh=Mask×weight+1

(3) loss= ∑ylogy ˆ+1−ylog1−y ˆ×Wh,

where Wh is a hard example weight and weight is a hyperparameter. According to the Eq. (2), compared with a cross entropy loss function, the hard example loss will increase because of the parameter weighting of the hard example, which makes the model more attentive to the hard example in the training process.

Random channel attention mechanism.

Overfitting is a common problem when training deep neural networks because of the lack of training data or the relatively simple network. To solve this problem, in addition to data augmentation, regularization is an effective method to alleviate overfitting, e.g., dropout (Krizhevsky, Sutskever & Hinton, 2017) or L2 regularization. In this paper, we propose a novel regularization algorithm where the feature channels are randomly weighted during the model training phase. Different from channel attention mechanisms that provide specific weights on the feature channel, the random channel attention (RCA) mechanism allocates different and random weights to each channel. The method is demonstrated in Fig. 4. Therefore, the robustness of the deep learning model is enhanced due to the nature of the randomness of the training process when compared to the dropout method, which involves randomly setting the output of each hidden neuron to zero with a certain probability. RCA is a soft method that involves only weighting the feature channel and ensures that a deep learning model is easier to train. The experiment in part 3 demonstrates that the speed of training by RCA converges faster than dropout and L2 regularization. Furthermore, the performance of the model trained by RCA is better than that of the model trained by other regularization methods.

Figure 4 Random channel attention mechanism.

Random weight values will be weighting to each channel of feature maps.

Double branch fusion U-Net

Experiments show that paying more attention to hard examples during the training process can improve segmentation results of a hard example region, however, this process will bring more false positive samples in the holistic segmentation result. To improve the performance of a hard example region segmentation, without increasing the false positive rate, we assume that the model is composed of a single encoder, two decoders, and a single fusion layer. The encoder is used to extract features from the original image, and the segmentation probability map generated by the decoder is based on features extracted by the encoder. One of the decoders is trained by cross entropy and weighted by hard example weights, while the other decoder is only trained by cross entropy. To fuse two decoder features, a fusion layer combines two branch decoders to generate the final segmentation result.

Inspired by U-net (Ronneberger & Brox, 2015), we propose the DBFU-Net with an overview architectural diagram as shown in Fig. 5. The network is composed of three parts that perform specific tasks: an encoder sub-network extracts high-level image features, two-decoder sub-networks generate a rough segmentation result, and one fusion layer combines features extracted by the two decoders to compute the final segmentation result. Each branch has a loss function to optimize all parameters. Like the deeply supervised training strategy, the proposed method will avoid the risk of increasing the network’s depth increasing the complexity of the optimization. The model’s focus needs to be on hard examples, therefore, one of the decoder branches is trained by the loss function (Eq. 3). The block used for the proposed DBFU-Net is improved by the res-block inspired by Link-Net (Chaurasia & Culurciello, 2017), which combines RCA to alleviate overfitting. The structure of the Res-block of the DBFU-Net is shown in Fig. 6.

Figure 5 The architecture of the proposed DBFU-Net.

DBFU-Net is composed of four parts: an encoder, the first decoder, the second decoder, and the fusion layer.

Figure 6 The architecture of Res-block in DBFU-Net.

Implementation details

We provide implementation details, which mainly includes preprocessing, training the first decoder, training the second decoder, training the fusion layer, and post processing. The detailed description of each step is listed as follows:

Preprocessing. To fit input data into the RVS model, we apply a preprocessing step to the fundus image. Because the blood vessels manifest high contrast in the green channel (Yin et al., 2015), we extract the green channel images, given an RGB fundus image. Since the network has a downsampling factor of 5, the size of the input image should be divisible by 25, therefore we had to pad the input image with multiples of 25. To adjust image contrast, we use contrast limited adaptive histogram equalization to enhance the input image. Then, we utilize a morphology method to obtain the hard example mask according to the label.

The lack of labeled data is one of the most difficult problems for RVS. Consider the DRIVE dataset as an example, the training set of the DRIVE dataset only contains 20 pictures. For the supervised algorithms, the use of data augmentation technology alleviates the problem of data scarcity. In this paper, we augment the training data using rotating, mirroring, and translating operations. Additionally, we use random elastic deformations to augment the training data to obtain more morphological characteristics of vessel. The process of random elastic deformation is shown in Fig. 7.

Figure 7 Augmented data by random elastic deformation.

Training the first decoder. We train the first decoder to obtain the parameter for the encoder. In this process, the learning rate is initially set to 7e−4 and multiplied by 1/3 every 1/3 epoch; the batch size is 2. The network model is trained for 12 epochs with an SGD optimizer, and the parameters are randomly initialized by he-normal (He et al., 2015).

Training the second decoder. To ensure the model focuses more on hard examples, we train the second decoder with cross entropy after weighting the hard example. In this process, we freeze the parameters of the encoder, and the parameters of the second decoder are randomly initialized by he-normal. The learning rate is initially set to 7e−4 and multiplied by 1/3 every 1/3 epoch, and the batch size is 2. The network model is trained for 12 epochs with an SGD optimizer. After that, we have fine-tuned all the parameters for 8 epochs with a learning rate of 5e−5.

Training the fusion layer. The first decoder can obtain fundus vessels from the features that are extracted by the encoder, but the second decoder focuses more on thin vessels. Therefore, we train the fusion layer to combine features from the two branches to obtain a segmentation result that is better than using only one branch. In this process, we freeze all parameters except fusion layers made of parameters randomly initialized by he-normal. The learning rate is initially set to 7e−4 and multiplied by 1/3 every 1/3 epoch; the batch size is 2. The network model is trained for 6 epochs with the SGD optimizer. After that, we fine-tune all parameters for 4 epochs with a learning rate of 5e−5.

Post processing. The range of segmentation probability map generated by model is [0, 1]. To ensure the segmentation result is in the form of gray images, we normalize the segmentation probability map to a range of [0, 255]. The final probability map is converted into binary images by applying the global threshold segmentation algorithm. Different segmentation performances will be achieved when applying different binarization thresholds. We choose the threshold that has the highest F1-score for the validation set as the optimal threshold value. The best threshold of the model varies due to different output results, which can reflect the best performance of the various models.

Results

In this section we will present our experimental datasets and settings, as well as the RVS performance indicator and experiment results.

Materials and experimental settings

Similar to most RVS work, we evaluated the proposed method using DRIVE (Digital Retinal Images for Vessel Extraction) (Staal et al., 2004), STARE (Structured Analysis of the Retina) (Fraz et al., 2012) and CHASE (Child Heart Health Study in England) (Hoover, Kouznetsova & Goldbaum, 2000) datasets, which are shown in Fig. 8. We find that different datasets have different data distribution characteristics. The DRIVE dataset contains 40 color images with a resolution of 565 × 584, which are captured at 45° field of view (FOV) and divided into a training set and a test set equally. The STARE dataset has 20 color fundus images that are captured at 35° FOV. The resolution of each image is 700 ×605. The CHASE dataset provides 14 paired color images with a resolution of 999 × 960. The images were collected from both the left and right eyes of school children. Note that in these datasets, each image has two manually labeled binary images with an FOV mask. We choose the binary images of the first observer as the ground truth. In the experiment using the DRIVE dataset, we tested model by the official test set. We perform the five-fold and four-fold cross-validation for the STARE and CHASE datasets because they have no official test datasets. In all experiments, we divided 10% of the training set as the validation set and select the model with the best performance in the validation set for testing to determine the threshold of binarization based on the selected model. The experimental computer has a Windows Server 2016 operating system running on two Intel Core Xeon Gold 6234 CPUs and two NVIDIA Tesla V100 Graphics Processing Units (GPUs).

Figure 8 Sample fundus images in DRIVE (rows (I)), STARE (rows (II)) and CHASE (rows (III)) dataset.

(A) Fundus image; (B) label of (A); (C) FOV mask of (A).

Performance measurements

RVS is a pixel-level binary classification problem in which each pixel can be divided as vessel and non-vessel pixels. The positive examples are the vessel pixels and the negative examples are non-vessel pixels. We can evaluate the RVS performance based on a confusion matrix that includes a TP (Ture Positive), TN (Ture Negative), FP (False Positive) and FN (False Negative). Then, based on the evaluation, we can generate the receiver operating characteristic (ROC) curve (Fawcett, 2006) to calculate the area under the ROC curve (AUC). In this paper, the RVS performance is measured by F1-score, sensitivity (Se), specificity (Sp), accuracy (Acc), G-mean, Matthews Correlation Coefficient (MCC) and AUC, which are defined as follows:

(4) F1Score=2×TP2×TP+FP+FN

(5) Sn=TPTP+FN

(6) Sp=TNTN+FP

(7) Acc=TP+TNTP+TN+FP+FN

(8) G−Mean=Sn×Sp

(9) MCC=TP×TNTP+TNTP+FNTN+FPTN+FN

The ratio of positive and negative examples is 1 to 9 according to the statistics of the data set. Therefore, the Acc will reach 90% but the Sn is 0 when all pixels are classified as negative examples. That is the reason why making ACC as the main evaluation indicator (Khanal & Estrada, 2019) is inappropriate. We should consider both Sn and Sp when measuring RVS performance because Sn and Sp only focuses on positive and negative examples. The MCC and F1-Score consider all categories of possible classification situations at the same time. Therefore, both the MCC and F1-Score can be used in the case of uneven samples; this model is commonly regarded as a balanced evaluation indicator. In this paper, all RVS indicators were calculated using only pixels inside FOVs over all the test images.

Experimental results

In this part, we conducted ablation experiments of DBFU-Net and show the performance of DBFU-Net on DRIVE, STARE and CHASE datasets.

Comparison with other regularization method

To compare the performance of different regularization methods, we used U-Net with res-block (single branch model) and DBFU-Net to compare the performance of model training, by different regularization methods and by using training models with no regularization method in three datasets. The comparison results on DRIVE, STARE and CHASE datasets are shown in Table 1. The dropout rate is set to 0.5, the L2 regularization parameter is set to 1e−3, the mean of weight is set to 1 and var is set to 0.5 of the RCA in all experiments. To show that the RCA can have better regularization capabilities on different models, we also used HR-Net (Sun, Liu & Wang, 2019) for comparative experiments.

Table 1 Performance of models tested on DRIVE, STARE and CHASE dataset. The top results for each dataset are marked in bold.

Dataset	Model	Regularization Method	Threshold	ACC	Se	Sp	F1	G	MCC	AUC	
DRIVE	Single branch	No	121	0.9521	0.8116	0.9704	0.8144	0.8874	0.7998	0.9789	
Dropout	117	0.9531	0.8185	0.9721	0.8202	0.8920	0.8001	0.9791	
L2	127	0.9550	0.8231	0.9745	0.8246	0.8956	0.8008	0.9799	
RCA	125	0.9559	0.8265	0.9762	0.8259	0.8982	0.8034	0.9805	
DBFU-Net	No	142	0.9412	0.8107	0.9711	0.8099	0.8873	0.7996	0.9786	
Dropout	139	0.9549	0.8199	0.9734	0.8221	0.8934	0.8012	0.9792	
L2	147	0.9558	0.8260	0.9762	0.8260	0.8980	0.8023	0.9800	
RCA	150	0.9563	0.8281	0.9771	0.8289	0.8995	0.8056	0.9811	
HR-Net	No	111	0.9530	0.8110	0.9711	0.8124	0.8874	0.7992	0.9782	
Dropout	122	0.9536	0.8179	0.9725	0.8210	0.8919	0.7999	0.9780	
L2	118	0.9552	0.8240	0.9743	0.8244	0.8960	0.8003	0.9793	
RCA	121	0.9558	0.8266	0.9759	0.8243	0.8982	0.8035	0.9801	
STARE	Single branch	No	140	0.9598	0.8231	0.9712	0.8215	0.8941	0.8204	0.9817	
Dropout	138	0.9615	0.8442	0.9764	0.8342	0.9079	0.8237	0.9831	
L2	151	0.9662	0.8519	0.9798	0.8428	0.9136	0.8273	0.9868	
RCA	142	0.9681	0.8598	0.9819	0.8493	0.9188	0.8297	0.9890	
DBFU-Net	No	147	0.9527	0.8162	0.9710	0.8213	0.8902	0.8211	0.9814	
Dropout	158	0.9653	0.8498	0.9782	0.8394	0.9117	0.8262	0.9860	
L2	164	0.9672	0.8527	0.9799	0.9437	0.9141	0.8301	0.9887	
RCA	163	0.9691	0.8612	0.9823	0.8501	0.9198	0.8332	0.9892	
HR-Net	No	136	0.9588	0.8221	0.9704	0.8211	0.8932	0.8199	0.9811	
Dropout	135	0.9606	0.8437	0.9760	0.8332	0.9074	0.8225	0.9823	
L2	144	0.9659	0.8502	0.9799	0.8414	0.9127	0.8264	0.9854	
RCA	147	0.9678	0.8588	0.9802	0.8487	0.9175	0.8290	0.9876	
CHASE	Single branch	No	130	0.9590	0.8391	0.9705	0.8217	0.9024	0.8097	0.9816	
Dropout	131	0.9611	0.8441	0.9765	0.8305	0.9079	0.8162	0.9854	
L2	140	0.9663	0.8489	0.9798	0.8357	0.9120	0.8149	0.9868	
RCA	138	0.9676	0.8507	0.9796	0.8363	0.9129	0.8181	0.9873	
DBFU-Net	No	142	0.9556	0.8372	0.9669	0.8204	0.8997	0.8091	0.9802	
Dropout	146	0.9635	0.8462	0.9781	0.8312	0.9101	0.8174	0.9861	
L2	166	0.9677	0.8508	0.9799	0.8364	0.9131	0.8169	0.9869	
RCA	161	0.9682	0.8520	0.9800	0.8375	0.9138	0.8195	0.9879	
HR-Net	No	125	0.9577	0.8372	0.9698	0.8211	0.9011	0.8086	0.9806	
Dropout	124	0.9601	0.8429	0.9745	0.8284	0.9063	0.8158	0.9849	
L2	138	0.9645	0.8470	0.9782	0.8350	0.9102	0.8141	0.9850	
RCA	133	0.9658	0.8499	0.9788	0.8348	0.9121	0.8172	0.9862	

Comparison of hard example weighting strategy

To verify the effectiveness of the hard example weighting strategy, we conducted a comparative experiment using DRIVE, STARE and CHASE datasets, to compare the performance of a single-branch model. The two-branch model does not use the hard example weighting training strategy (DBFU-Net-NH), decoder 1 of DBFU-Net, decoder 2 of DBFU-Net or the DBFU-Net. The experimental results are shown in Table 2. The second branch of DBFU-Net-NH was trained by cross entropy. All controlled experiments in this section use the RCA regularization method.

Table 2 Performance of the five models tested on the DRIVE, STARE and CHASE datasets.

The top results for each dataset are marked in bold. DBFU-Net-NH is shorthand for: decoder 2 of DBFU-Net is trained by cross entropy and not using the hard example weighting training strategy.

Dataset	Method	Threshold	ACC	Se	Sp	F1	G	MCC	AUC	
DRIVE	single branch	125	0.9559	0.8265	0.9762	0.8259	0.8982	0.8034	0.9805	
DBFU-Net-NH	131	0.9554	0.8266	0.9770	0.8264	0.8987	0.8041	0.9801	
Decoder 1	122	0.9556	0.8266	0.9771	0.8257	0.8987	0.8033	0.9802	
Decoder 2	197	0.9497	0.8534	0.9141	0.8230	0.8832	0.8012	0.9768	
DBFU-Net	150	0.9563	0.8281	0.9771	0.8289	0.8995	0.8056	0.9811	
STARE	single branch	142	0.9681	0.8598	0.9819	0.8493	0.9188	0.8297	0.9890	
DBFU-Net-NH	145	0.9684	0.8599	0.9825	0.8494	0.9192	0.8312	0.9889	
Decoder 1	144	0.9679	0.8596	0.9820	0.8492	0.9188	0.8299	0.9887	
Decoder 2	206	0.9513	0.8976	0.9284	0.8355	0.9129	0.8203	0.9773	
DBFU-Net	159	0.9691	0.8612	0.9823	0.8501	0.9198	0.8332	0.9892	
CHASE	single branch	138	0.9676	0.8507	0.9796	0.8363	0.9129	0.8181	0.9873	
DBFU-Net-NH	140	0.9677	0.8510	0.9798	0.8370	0.9131	0.8183	0.9867	
Decoder 1	136	0.9673	0.8501	0.9789	0.8358	0.9122	0.8182	0.9868	
Decoder 2	215	0.9509	0.8937	0.9204	0.8211	0.9070	0.8099	0.9748	
DBFU-Net	161	0.9682	0.8520	0.9800	0.8375	0.9138	0.8195	0.9879	

Comparison with dice loss, focal loss

The second decoder of DBFU-Net can focus on the hard example in the training process. Focal loss (Lin et al., 2017) can also pay more attention to hard example pixels, shown in Eq. (9). Dice loss (Milletari, Navab & Ahmadi, 2016) is proposed for uneven data distributions; the effect of focal loss and dice loss in the training process are like the proposed training strategy, which weights hard examples as shown in Eq. (11). Therefore, we compared the proposed hard example weighting strategy with focal loss and dice loss. The parameter γ of focal loss was set to 2 and the parameter ɛ of dice loss was set to 1e−5. The second decoder was trained by the hard example weighting strategy, focal loss, and dice loss. Then we used the result of the fusion layer as the final comparative result. In addition, we compared the performance of the second decoder of DBFU-Net and the single branch model that was trained by focal loss and dice loss. The comparative experiment results for the DRIVE, STARE and CHASE datasets are shown in Table 3. SB-F represents a single branch model trained by focal loss; SB-D represents a single branch model trained by dice loss. DBFU-Net-F represents a DBFU-Net trained by focal loss. DBFU-Net-D represents a DBFU-Net trained by dice loss; decoder 2 represents a second decoder of DBFU-Net. All controlled experiments in this section use an RCA regularization method. (10) FLpt=−1−ptγ logpt

(11) pt=p,ify=11−p,otherwise,

Table 3 Performance of the seven models tested on the DRIVE, STARE and CHASE dataset.

The top results for each dataset are marked in bold. SB-F represents a single branch model trained by focal loss, SB-D represents a single branch model trained by dice.

Dataset	Method	Threshold	ACC	Se	Sp	F1	G	MCC	AUC	
DRIVE	DBFU-Net-F	162	0.9560	0.8271	0.9765	0.8278	0.8987	0.8044	0.9809	
DBFU-Net-D	141	0.9557	0.8259	0.9769	0.8270	0.8982	0.8038	0.9802	
SB-F	168	0.9513	0.8413	0.9413	0.8190	0.8900	0.8001	0.9764	
SB-D	147	0.9542	0.8347	0.9568	0.8239	0.8937	0.8013	0.9789	
Decoder 2	197	0.9497	0.8534	0.9141	0.8230	0.8832	0.8012	0.9768	
DBFU-Net	150	0.9563	0.8281	0.9771	0.8289	0.8995	0.8056	0.9811	
STARE	DBFU-Net-F	169	0.9679	0.8603	0.9808	0.8495	0.9185	0.8321	0.9891	
DBFU-Net-D	152	0.9689	0.8591	0.9810	0.8490	0.9180	0.8316	0.9890	
SB-F	174	0.9544	0.8839	0.9325	0.8402	0.9079	0.8227	0.9817	
SB-D	155	0.9596	0.8783	0.9457	0.8429	0.9114	0.8275	0.9877	
Decoder 2	206	0.9513	0.8976	0.9284	0.8355	0.9129	0.8203	0.9773	
DBFU-Net	163	0.9691	0.8612	0.9823	0.8501	0.9198	0.8332	0.9892	
CHASE	DBFU-Net-F	197	0.9667	0.8512	0.9787	0.8369	0.9127	0.8186	0.9875	
DBFU-Net-D	188	0.9672	0.8501	0.9804	0.8358	0.9129	0.8181	0.9869	
SB-F	195	0.9518	0.8902	0.9299	0.8189	0.9098	0.8003	0.9716	
SB-D	176	0.9588	0.8740	0.9411	0.8245	0.9069	0.8107	0.9823	
Decoder 2	215	0.9509	0.8937	0.9204	0.8211	0.9070	0.8099	0.9748	
DBFU-Net	161	0.9682	0.8520	0.9800	0.8375	0.9138	0.8195	0.9879	

where γ is a hyperparameter, p is output possibility of deep learning model, y is label.

(12) Diceloss=1−2I+ɛU+ɛ

(13) I= ∑tiyi

(14) U= ∑ti+ ∑yi

where ti is label, yi is output possibility of deep learning model, ɛ is a hyperparameter.

Discussion

This section analyzes the results of the ablation experiment of DBFU-Net. We analyzed the effect of RCA and hard example weighting training strategy on the ablation experiment. We also compared the performance of DBFU-Net to other published methods.

Comparison with other regularization method

It is easy to overfit when training the RVS deep learning model, that is why we needed to come up with an effective regularization method. The proposed RCA is an effective regularization method, shown in Fig. 9, which compares it to the Drop out and L2 regularization training curves.

Figure 9 Training loss curves of the model trained by different regularization methods using DRIVE, STARE and CHASE datasets.

The columns of (a), (b), and (c) represent the training process performance of the model using the DRIVE, STARE, and CHASE data sets, respectively. Row (I) is the training loss curve of single branch model for each dataset; row (II) is the training loss curve of DBFU-Net for each dataset; row (III) is the training loss curve of HR-Net for each dataset.

Each regularization method shown in Table 1, Figs. 9 and 10, performs similarly on the different datasets. Compared with a single branch model, the validation loss of DBFU-Net converges more slowly because DBFU-Net has more parameters. In the blank contrast group of a single branch model and DBFU-Net method without any regularization method, training loss can always converge given an increase in the count of the iteration; validation loss nevertheless quickly rises after a certain degree of convergence, that is an obvious overfitting phenomenon. Training loss and validation loss can still maintain a stable convergence state when iterating for a long period of time in the dropout experimental ground. However, the stable convergence value of validation loss is at a relatively high level, so the segmentation effect is not well. In the L2 regularization experimental group, the validation loss can converge steadily and concurrent with the training loss at a stable convergence state. But the validation loss will rise and the training loss will converge after more experiment iterations, that is the phenomenon of overfitting. Proposing the RCA regularization method can ensure that training loss and validation loss converge rapidly and at the same time. The validation loss can maintain a steady state with increased train steps. The best validation loss of a model trained by RCA is to lower the models trained by other regularization methods. From Table 1, the segmentation performance of the model trained by RCA is better than that for the other methods in the three datasets. In addition, we found that the HR-Net trained by RCA can obtain better performance than other regularization methods. Therefore, we can draw the conclusion that the proposed RCA regularization method is better than other regularization methods.

Figure 10 Validation loss curves of the model trained by different regularization methods using DRIVE, STARE and CHASE datasets.

The columns of (A), (B), and (C) represent the training process performance of the model for the DRIVE, STARE, and CHASE data sets, respectively. The row (I) is the validation loss curve of a single branch model for each dataset; row (II) is the validation loss curve of DBFU-Net for each dataset; row (III) is the validation loss curve of HR-Net for each dataset.

The effective of hard example weighting training strategy

According to Table 2, the segmentation performance of DBFU-Net ranks first. The best threshold of the second decoder is the one that is higher than the contrasting result. Because the second decoder was trained using a hard example weighting strategy, the decoder paid more attention to the areas that could be a vessel, which improved the segmentation recall result, but adds more false positive points. To obtain a better segmentation result for the comprehensive performance index F1-Score, a higher threshold is required to filter the false positive points. Although the performance of the DBFU-Net-NH is worse than that of DBFU-Net, it is better than that of a model with a single branch; this is generally because DBFU-Net-NH contains more parameters. Therefore, we can draw the conclusion that a hard example weighting training strategy can improve segmentation performance. The output probability distribution maps of the first decoder, second decoder, and final fusion layer is shown in Fig. 11. The contrast of the final fusion output probability map and output of the double decoders is shown Fig. 12.

Figure 11 Performance of DBFU-Net given three datasets.

Each row of figure represents the performance of DBFU-Net on the DRIVE (A–E), STARE (F–J), and CHASE (K–O) data sets, respectively. Each column of figure represents the original fundus images (A–K); the label of corresponding original fundus images (B–L); the outputs of first decoder (C–M); the outputs of second decoder (D–N); the outputs of fusion layer and that are final segmentation probability distribution maps (E–Q), respectively.

Figure 12 The contrast of the final fusion output probability map and output of double decoders.

Each row of figure represents the DRIVE (A–F), STARE (G–L), and CHASE (M–R) data sets, respectively. Each column of figure represents the original fundus images (A–M); the detail of the regions (B–N) that have been selected in corresponding original fundus images; the labels of corresponding regions (C-O); the final segmentation results generated by DBFU-Net (D–P); the segmentation results of second decoder (E–Q); the segmentation results of the first decoder (F–R). In the final segmentation results, the segmentation results of second decoder, and the segmentation results of the first decoder, we used green, blue, red and black to represent TP, FP, FN, TN, respectively.

According to Figs. 11 and 12, DBFU-Net detects more positive examples than the model with a single decoder. It reduces the false positive rate when compared with the second decoder. In other words, DBFU-Net can combine the advantages of the first decoder and the second decoder, reducing the impact of their respective shortcomings.

Focal loss also pays more attention to hard example pixels; Dice loss is proposed for the uneven data distribution. The effect of focal loss and dice loss in the training process is like the proposed training strategy. Thus, we can compare the hard example weighting training strategy with focal loss and dice loss. According to Table 3, we found that the performance of the DBFU-Net trained by a hard example weighting training strategy is better than other methods, especially in terms of recall. Therefore, we can draw a conclusion that the ability of attention on the hard example of the proposed method is better than that of focal loss and dice loss. In addition, the recall and the best of threshold of a single branch model was trained by focal loss and the second decoder of the DBFU-Net are significantly higher than other control results because the focal loss and hard example weighting training strategy can pay attention to hard example area, but the recall of the second decoder of DBFU-Net is higher than that of the single branch model trained by focal loss. Therefore, we can draw the conclusion that using hard example weighting training strategy can pay attention to hard example area more efficiently than focal loss in RVS.

Comparison against existing methods

As shown in Tables 4, 5 and 6, we compared the proposed method with those of state-of-the-art methods using the DRIVE, STARE and CHASE datasets, A dash (-) indicates that the values are not given in these papers. DBFU-Net performs the best among those methods in terms of the F1-score, Sn, G-mean and MCC, which indicates that when compared to other approaches, the DBFU-Net shows state-of-the-art efficiency. The ACC is the third highest for the DRIVE dataset. Even though our approach performs marginally worse than other methods in terms of Sp, it significantly outperforms these methods from the view of other metrics, especially for the F1-score, which is considered as the primary metric in RVS. Additionally, Acc and Sp are considered reference indicators due to the nature of one-sidedness. Therefore, we can conclude that our proposed method is superior to other methods. According to Table 4, Table 5 and Table 6, the proposed DBFU-Net achieves state-of-the-art performance for the three datasets tested. Fig. 13 shows the performance of the method on hard examples. The image shows that in the optic disc area, our method avoids the situation where the edge of the optic disc is predicted to be a blood vessel. Our method shows a better segmentation performance than other methods in the small vascular area. Additionally, our segmentation results were not affected by retinal spots and have obtained lower FP in the lesion area when compared to the segmentation performances of other methods.

Table 4 Performance of the different models tested on the DRIVE dataset.

The top results for each dataset are marked in bold. A dash (-) indicates that the values are not given in these papers.

Type	Method	ACC	Se	Sp	F1	G	MCC	AUC	
non-learning	Sazak, Nelson & Obara (2019)	0.959	0.718	0.981	–	0.8393	-	-	
unsupervised	Câmara Neto et al. (2017)	0.8718	0.7806	0.9629	–	0.867	–	–	
Zhang et al. (2020)	0.959	0.745	0.976	–	0.8527	–	0.861	
supervised	Yang et al. (2020)	0.9694	0.7762	0.9835	–	0.8737	–	–	
Leopold et al. (2017)	0.9106	0.6963	0.9573	0.7382	0.8159	0.6820	–	
Li et al. (2020)	0.9557	0.7890	0.9799	0.8192	0.8793	–	0.9774	
Kromm & Rohr (2020)	0.9547	0.7651	0.9818	0.8192	0.8667	–	0.9750	
Ribeiro, Lopes & Silva (2019)	0.9569	0.7880	0.9819	–	0.8796	–	–	
Adarsh et al. (2020)	0.9563	0.7979	0.9794	0.8227	0.8840	–	0.9795	
Guo et al. (2019)	0.9561	0.7891	0.9804	0.8249	0.8796	0.7964	0.9806	
Yan, Yang & Cheng (2019)	0.9538	0.7631	0.9820	-	0.8657	-	0.9750	
Palanivel, Natarajan & Gopalakrishnan (2020)	0.9480	0.7375	0.9788	-	0.8496	-	0.9590	
Zhao, Li & Cheng (2020)	–	0.8329	0.9767	0.8229	0.9019	-	-	
Li, Comer & Zerubia (2020)	0.9339	0.8063	0.9529	–	0.8761	-	-	
DBFU-Net	0.9563	0.8281	0.9771	0.8289	0.8995	0.8056	0.9811	

Table 5 Performance of the different models tested on the STARE dataset.

The top results for each dataset are marked in bold. A dash (-) indicates that the values are not given in these papers.

Type	Method	ACC	Se	Sp	F1	G	MCC	AUC	
non-learning	Sazak, Nelson & Obara (2019)	0.962	0.730	0.979	–	0.8453	–	–	
unsupervised	Câmara Neto et al. (2017)	0.8894	0.8344	0.9443	–	0.8877	–	–	
Zhang et al. (2020)	0.951	0.799	0.962	–	0.8767	–		
supervised	Yang et al. (2020)	0.9537	0.7721	0.9885	–	0.8736	–	–	
Leopold et al. (2017)	0.9045	0.6433	0.9472	0.6465	0.7797	0.5960	–	
Li et al. (2020)	0.9620	0.7798	0.9822	0.8037	0.8752	–	0.9791	
Guo et al. (2019)	0.9674	0.8212	0.9843	0.8421	0.8991	0.8221	0.9859	
Yan, Yang & Cheng (2019)	0.9638	0.7735	0.9857	–	0.8732	–	0.9872	
Palanivel, Natarajan & Gopalakrishnan (2020)	0.9542	0.7484	0.9780	–	0.8555	–	0.9711	
Zhao, Li & Cheng (2020)	–	0.6433	0.9472	0.6465	0.7806	–	–	
Li, Comer & Zerubia (2020)	0.9422	0.8394	0.9536	–	0.8932	–	–	
DBFU-Net	0.9691	0.8612	0.9823	0.8501	0.9198	0.8332	0.9892	

Table 6 Performance of the different models tested on the CHASE dataset.

The top results for each dataset are marked in bold. A dash (-) indicates that the values are not given in these papers.

Type	Method	ACC	Se	Sp	F1	G	MCC	AUC	
non-learning	Sazak, Nelson & Obara (2019)	0.9630	0.831	0.9810	–	0.9028	–	–	
supervised	Leopold et al. (2017)	0.8936	0.8618	0.8961	0.5391	0.8787	0.5376	–	
Li et al. (2020)	0.9581	0.7536	0.9808	0.7826	0.8597	–	0.9721	
Guo et al. (2019)	0.9627	0.7888	0.9801	0.7983	0.8793	0.7733	0.9840	
Yan, Yang & Cheng (2019)	0.9607	0.7641	0.9806	–	0.8656	–	0.9840	
Zhao, Li & Cheng (2020)	–	0.8618	0.8961	0.5391	0.8788	–	–	
DBFU-Net	0.9682	0.8520	0.9800	0.8375	0.9138	0.8195	0.9879	

Figure 13 The performances of different methods in hard examples.

Each row of figure shows detection of optic discs (A–F), small vascular (G–L), and lesion areas (M–R), respectively. Each column of figure shows the original images (A–M), the ground truths of corresponding original images (B–N), outputs by Fu et al. (2016) (C–O), Orlando, Prokofyeva & Blaschko (2017) (D–P), Niemeijer et al. (2004) (E–Q) and our method (F–R), respectively. The places where are highlighted in original images respectively show the focused areas.

Table 7 Model performance measures from cross-training.

Our result from model that without fine tuning. The top results for each dataset are marked in bold. DBFU-Net-Dropout represents a DBFU-Net trained by DropoutDBFU-Net-L2 represents a DBFU-Net trained by L2 Regularization. A dash (-) indicates that the values are not given in these papers.

Test on	Train on	Method	ACC	Se	Sp	F1	G	MCC	AUC	
DRIVE	STARE	Leopold et al. (2017)	0.8748	0.5110	0.9533	0.5907	0.6974	0.5309	0.7322	
Guo et al. (2019)	0.9502	0.7446	0.9784	–	0.8535	–	0.9709	
Yan, Yang & Cheng (2019)	0.9444	0.7014	0.9802	–	0.8292	–	0.9568	
single branch-Dropout	0.9411	0.7611	0.9589	0.7659	0.8543	0.7496	0.9492	
single branch-L2	0.9421	0.7869	0.9643	0.7799	0.8711	0.9531	0.9528	
single branch	0.9428	0.7884	0.9656	0.7815	0.8725	0.7548	0.9537	
DBFU-Net-Dropout	0.9423	0.7914	0.9645	0.7802	0.8737	0.7519	0.9521	
DBFU-Net-L2	0.9475	0.7932	0.9667	0.7923	0.8757	0.7603	0.9542	
DBFU-Net	0.9532	0.8046	0.9712	0.8091	0.8840	0.7851	0.9570	
CHASE	Leopold et al. (2017)	0.8796	0.6222	0.9355	0.6463	0.7622	0.5768	0.7788	
Guo et al. (2019)	0.9377	0.6960	0.9699	–	0.8216	–	0.9523	
single branch-Dropout	0.9274	0.7211	0.9418	0.7653	0.8241	0.7495	0.9386	
single branch-L2	0.9313	0.7352	0.9506	0.7827	0.8360	0.7609	0.9414	
single branch	0.9334	0.7435	0.9512	0.7894	0.8410	0.7643	0.9432	
DBFU-Net-Dropout	0.9310	0.7522	0.9498	0.7803	0.8452	0.7597	0.9413	
DBFU-Net-L2	0.9342	0.7833	0.9517	0.7964	0.8634	0.7832	0.9516	
DBFU-Net	0.9379	0.7984	0.9596	0.8072	0.8753	0.7895	0.9556	
STARE	DRIVE	Leopold et al. (2017)	0.9070	0.7842	0.9265	0.6916	0.8519	0.6465	0.8533	
Guo et al. (2019)	0.9548	0.7188	0.9816	–	0.8400	–	0.9686	
Yan, Yang & Cheng (2019)	0.9580	0.7319	0.9840	–	0.8486	–	0.9678	
single branch-Dropout	0.9403	0.7857	0.9685	0.7882	0.8723	0.7709	0.9508	
single branch-L2	0.9496	0.7923	0.9702	0.7923	0.8767	0.7768	0.9542	
single branch	0.9512	0.8097	0.9711	0.7999	0.8867	0.7814	0.9586	
DBFU-Net-Dropout	0.9505	0.8132	0.9707	0.7962	0.8885	0.7770	0.9578	
DBFU-Net-L2	0.9562	0.8201	0.9744	0.8087	0.8939	0.7892	0.9601	
DBFU-Net	0.9601	0.8235	0.9761	0.8121	0.8966	0.7935	0.9701	
CHASE	Leopold et al. (2017)	0.8771	0.6973	0.9062	0.6057	0.7941	0.5441	0.8017	
Guo et al. (2019)	0.9501	0.6799	0.9808	–	0.8166	–	0.9686	
single branch-Dropout	0.9413	0.7724	0.9597	0.7743	0.8610	0.7522	0.9502	
single branch-L2	0.9485	0.7813	0.9624	0.7862	0.8671	0.7597	0.9563	
single branch	0.9502	0.7845	0.9689	0.7894	0.8718	0.7610	0.9598	
DBFU-Net-Dropout	0.9495	0.7902	0.9603	0.7884	0.8711	0.7603	0.9592	
DBFU-Net-L2	0.9537	0.7993	0.9677	0.7921	0.8795	0.7654	0.9615	
DBFU-Net	0.9565	0.8045	0.9704	0.8017	0.8836	0.7756	0.9691	
CHASE	DRIVE	Leopold et al. (2017)	0.8901	0.9038	0.8891	0.5416	0.8963	0.5480	0.8964	
Guo et al. (2019)	0.9411	0.6726	0.9710	–	0.8081	–	0.9511	
single branch-Dropout	0.9302	0.8299	0.9204	0.8048	0.8740	0.7842	0.9478	
single branch-L2	0.9375	0.8315	0.9255	0.8097	0.8772	0.7891	0.9486	
single branch	0.9387	0.8312	0.9298	0.8102	0.8791	0.7903	0.9509	
DBFU-Net-Dropout	0.9382	0.8347	0.9287	0.8104	0.8804	0.7898	0.9507	
DBFU-Net-L2	0.9425	0.8386	0.9304	0.8148	0.8833	0.7962	0.9568	
DBFU-Net	0.9457	0.8404	0.9312	0.8198	0.8846	0.7991	0.9638	
STARE	Leopold et al. (2017)	0.9173	0.7525	0.9302	0.5688	0.8365	0.5475	0.8413	
Guo et al. (2019)	0.9441	0.6980	0.9715	–	0.8235	–	0.9565	
single branch-Dropout	0.9327	0.7962	0.9439	0.8084	0.8669	0.7808	0.9571	
single branch-L2	0.9358	0.8013	0.9482	0.8102	0.8717	0.7843	0.9593	
single branch	0.9395	0.8024	0.9497	0.8106	0.8729	0.7865	0.9602	
DBFU-Net-Dropout	0.9364	0.8089	0.9473	0.8104	0.8754	0.7853	0.9611	
DBFU-Net-L2	0.9411	0.8102	0.9501	0.8155	0.8774	0.7912	0.9645	
DBFU-Net	0.9466	0.8141	0.9539	0.8173	0.8812	0.7973	0.9692	

Cross-training experiment

The cross-training experiments reflects the robust performance of the proposed model in realistic situations (Zhou et al., 2017). Models with good robust performance can be applied to many realistic situations. The statistical results of the cross-training experiment using the three datasets are shown in Table 7, A dash (-) indicates that the values are not given in these papers. Compared with other methods, the proposed method had obtained the highest ACC, Se, F1 Score, G-Mean, MCC and AUC. The cross-training experiment not only showed that the proposed method can be applied to real-world situations and reflect that the robustness of DBFU-Net is better than that of a single branch model. In addition, in the experimental group of DBFU-Net and a single branch, the robust performance of the model trained via RCA regularization is better than the model trained by L2 regularization or drop out. Because the model was randomly introduced when using RCA training, the robustness of the model was able to be increased.

Conclusions and feature work

This paper aims at proposing a novel deep learning architecture, DBFU-Net, to segment retinal vessels. To avoid overfitting, we propose to apply RCA and to randomly weight each feature map channel. Hard example masks were introduced to guide the model to pay more attention to the edge of large vessels and thin vessel areas. To reduce the computational cost of extracting the hard example mask, we propose a novel hard example extraction method based on morphology. The experiment proved that the second training decoder achieves a performance gain when weighting hard examples due to the loss function. Our proposed method also obtained state-of-the-art results for DRIVE, STARE and CHASE dataset.

We plan to examine two additional aspects in the future. First, hard example weighting is proven to be effective for RVS. We will use this method and combine it with other segmentation models for other segmentation tasks as well. DBFU-Net is a double branch model that is composed of 4 parts. Moreover, the computational cost of the hard example weighting strategy is greater than that of focal loss and dice loss because morphological operations bring additional computational costs. Hence, we will explore new methods that are less computationally expensive or based on hard example weighting training strategy.

Supplemental Information

Supplemental Information 1 Experimental code that contained preprocess script and DBFU-Net model python code

Click here for additional data file.

Additional Information and Declarations

Competing Interests

Author Contributions

Data Availability

The authors declare there are no competing interests.

Jianping Huang and Zefang Lin conceived and designed the experiments, performed the experiments, analyzed the data, performed the computation work, prepared figures and/or tables, and approved the final draft.

Yingyin Chen and Xiao Zhang performed the experiments, prepared figures and/or tables, and approved the final draft.

Wei Zhao and Jie Zhang analyzed the data, prepared figures and/or tables, and approved the final draft.

Yong Li performed the computation work, authored or reviewed drafts of the paper, and approved the final draft.

Xu He and Meixiao Zhan performed the experiments, authored or reviewed drafts of the paper, and approved the final draft.

Ligong Lu conceived and designed the experiments, performed the experiments, authored or reviewed drafts of the paper, and approved the final draft.

Xiaofei Jiang and Yongjun Peng conceived and designed the experiments, performed the computation work, authored or reviewed drafts of the paper, and approved the final draft.

The following information was supplied regarding data availability:

All experimental code (includeing hte preprocess code and DBFU-Net model code) is available in the Supplemental File.

The datasets used are as follows:

- DRIVE: http://www.isi.uu.nl/Research/Databases/DRIVE/

- STARE: http://cecas.clemson.edu/ ahoover/stare/

- CHASE: https://blogs.kingston.ac.uk/retinal/chasedb1/.

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
