# Peer review of "DBFU-Net: Double branch fusion U-Net with hard example weighting train strategy to segment retinal vessel"

_PeerJ Computer Science, doi:10.7717/peerj-cs.871_

## Round 0.1 · original submission · Major Revisions

The authors should improve the manuscript in many aspects.

Clarification in presentation
1. The comparisons with other methods are not clear. The authors should add more details in the experimental tables, e.g., Table 4-6 with the categorization of non-learning/learning methods, and their network architectures/parameter sizes.

2. The authors should polish the paper and fix all the grammar errors and typos for a better presentation.

Experimental setting
1. The authors should state clearly the training/test split and whether it follows the evaluation metric in other methods (e.g., choosing the threshold for the F-1 score).

2. During the evaluation, the author should have a validation set to find proper parameters, e.g., threshold for the F-1 score.

Experimental results
1. The single U-Net is based on Res-blocks, which already achieves a good baseline, while the proposed modules provide marginal improvements. It would be more interesting to conduct experiments based on similar network capacity to other deep-learning based methods, e.g., [20].

2. The authors should verify whether there is a need of using the two-branch model, e.g., reporting results of individual branches and providing more ablation studies.

3. It would be more informative to see the ablation study of proposed modules in the cross-training setting, which may potentially provide more improvement.

Reviewer 1 ·

Basic reporting

- The english writing is extremely problematic and hard to follow, the grammatical errors are all over the article. The submission should be thoroughly refined and proofread.
- The description in the section of related works is not informative enough to outline the pros and cons of different approaches and their design choices, thus being hard to well clarify the contribution of the proposed method.

Experimental design

- The train/test split is not well reported in the experimental settings. There are only descriptions on the cross-validation, but we have no idea what are the number of testing samples in each dataset. The authors should better clarify if they follow exactly the same experimental settings as other baselines used for making comparison.

Validity of the findings

- Although the reported performance of the proposed method is superior to various baselines on three datasets (i.e. DRIVE, STARE, an CHASE), the improvement is actually quite marginal. The ablation study for verifying the contribution of design choices (e.g. random channel attention, hard example weighting) shows also the same problem to have only incremental (almost neglectable) improvement with respect other model variants.
- The authors should clarify if they adopt the same binarization threshold for all the baselines while using F1 score, Sn, Sp, ACC, G-Mean, and MCC metrics for evaluation. As now the proposed method "chooses the threshold that has the highest F1-score as the optimal threshold value" (stated in Line 281), it would be inappropriate and unfair since the threshold is chosen according to the evaluation result (based on testing data?).

Additional comments

This paper proposes to tackle the problem of segmenting retinal vessel from the fundus image. The proposed method is stemmed from the U-Net architecture and extended to consist of one encoder and two decoder branches, where one encoder is trained with the typical cross-entropy objective while another one is trained by the loss function that weights more on the hard pixels (e.g. the boundary pixels of vessel). Moreover, the authors propose a "random channel attention" mechanism serving as regularization to improve the model training. The output of two decoder branches are merged by a fusion layer to produce the final segmentation output.

Pros:
+ The idea of adopting the hard example/pixels is not new but shown to be beneficial for the task of retinal vessel segmentation. The regularization based on "random channel attention" mechanism also shows better (but quite marginal) performance.
Cons:
In addition to the issues as commented in "Basic reporting", "Experimental design", and "Validity of the findings", here comes more problems that the authors should take into consideration for improving their paper.
- The performance of each decoder branch should be reported in order to better distinguish between the boost contributed to fusion layer and the hard example/pixels.
- During the ablation study for making comparison with respect to focal loss and dice loss, the parameter settings for the focal loss should be clarified.
- The superior performance in comparison to other baselines seems to mostly stem from the network with larger capacity, as the model variant of using single branch (in Table 2) already outperforms many baselines. There should be more detailed discussion/comparison on the model capacity among the proposed method and baselines for better clarifying the contribution of the proposed method.

Overall, as there exists the issues of having marginal improvement, unclear and potentially problematic experimental settings, insufficient experimental results, and bad paper organization/writing, I regret to suggest the rejection for this submission.

Reviewer 2 ·

Basic reporting

The paper proposed an automatic fundus image retinal vessel segmentation (RVS) method, consisting of a 2-branch fusion U-Net with the proposed random channel weighting regularization mechanism. While a branch was trained with segmentation criterion, the other was trained with morphological mask weighted criterion. Related works were categorized and compared with the proposed methods in reasonable manners. Figures about different parts of proposed method were shown with clarity.

However, the experiments demonstrated no significant proof that the proposed methods (regularization and weighting loss) performed better than the existing deep learning methods. Also, there were a lot of gramatically errors that need to be fixed.

Experimental design

The experiments consisted of baselines and ablation comparisons on 3 different datasets, which were techincally sound and adequate. Also, the description of the method and all settings of experiments were provided with sufficient details.

Validity of the findings

The proposed method should be competitively effective, but better not to be claimed superior to the exsisting methods. According the the results, the improvements of DBFU-Net over other regularization methods(L2, Dropout, etc), training losses (Dice, Focal), and existing deep learning segmentation baselines were quantitatively neglegable (<1%). Moreover, the necessity of two branches were kind of sceptical, as applying mask weighting loss with smaller "weight" in Eq. 3 on a single U-Net could be equal to the effect of fusing two branches. Last but not least, the computational cost of generating "morphological mask weighted cross entropy loss" should be greater than Focal and Dice loss because of the additional erosion/dilation operations.

Additional comments

The authors addressed a good problem to solve, as automatic RVS should be beneficial for ophthalmology disease screening. A fairly amount of experiments were conducted to show the performance of the proposed method. However, the following are a few cons to be fixed:

1. According to the results, the performance and improvement of the proposed methods over other exisiting methods should be claimed conservative.
2. Additional experiments should be conducted to better demonstrate the contribution of this work:
2.1. single branch U-Net vs. DBFU-Net, using the same loss functions (mask weighted, Focal, DICE)
2.2. single branch U-Net vs. DBFU-Net, using the same regularizations (RCA, dropout, etc.)
3. There were too many grammatical errors. I strongly suggest the authors proofread it thoroughly, e.g.
1.1. Missing “a” and “the”
1.2. Wrong usage of comma and multiple verbs in a sentense
1.4. Fig. 2, Func. 3
1.5. Typos, e.g.
Line 61. patient -> procedure
Line 188. define -> definition
Line 424. feature -> future

---

## Round 0.2 · Minor Revisions

Thanks for the authors' major revision of the manuscript, which addresses most issues raised by the reviewers. Therefore, I would recommend a minor revision, and please address other minor issues mentioned below and the ones in the 2nd-round reviews. Btw, to encourage reproducible research, the authors are highly encouraged to release the code and models to the public.

Regarding the concern of performance improvement raised by the reviewers, the authors should add a few sentences to discuss it in the paper.

It is better to describe more details in the caption of each table, e.g., the notation of each baseline like SB-F in Table 3 (and similarly for other tables).

The paper still requires a significant effort to fix the typos and grammar issues. To name a few,
Ln 325: they no official...
Ln 359: to "compare" the
Ln 360: models training models with no regularization method...
Ln 366: to "compare" the
Ln 406: "Compared" with
Fig. 1: "extracted"
Fig 4. Radom Weight (on the figure) -> Random

Reviewer 1 ·

Basic reporting

- The english writing is still problematic as the grammatical errors are still all over the article even in the revision parts. The submission should be thoroughly refined and proofread.
- The categorization in the section of related works seems awkward, the boundary between non-learning-based methods and some unsupervised methods is unclear.

Experimental design

no comment

Validity of the findings

- If the authors claim that their design choices (e.g. random channel attention, hard example weighting) can be combined with the techniques proposed in other articles to get better results, there should be such experiments in the revision.

Additional comments

This paper proposes to tackle the problem of segmenting retinal vessel from the fundus image. The proposed method is stemmed from the U-Net architecture and extended to consist of one encoder and two decoder branches, where one encoder is trained with the typical cross-entropy objective while another one is trained by the loss function that weights more on the hard pixels (e.g. the boundary pixels of vessel). Moreover, the authors propose a "random channel attention" mechanism serving as regularization to improve the model training. The output of two decoder branches are merged by a fusion layer to produce the final segmentation output.

Pros:
+ The idea of adopting the hard example/pixels is not new but shown to be beneficial for the task of retinal vessel segmentation. The regularization based on "random channel attention" mechanism also shows better (but quite marginal) performance.
Cons:
Please refer to my comments on "Basic reporting" and "Validity of the findings".

Overall, I suggest the major revision for this submission.

Reviewer 2 ·

Basic reporting

no comment

Experimental design

no comment

Validity of the findings

no comment

Additional comments

1. The author had clarified the incremental contributions of this paper: RCA regularization, hard example weighting train strategy, and DBFU-Net. Also, the results were claimed to be comparable with existing methods.

2. The paper had been proofread.

3. Additional experiments were conducted and the results were demonstrated to be competitive to the existing methods. A follow-up concern is that your tables got some "-" (missing data). I suggest the author fill them up as some might alter the expectation (e.g. Table 7 [21, 22] might be better than the proposed method).

---

## Round 0.3 · Minor Revisions

Overall, the authors have addressed all the concerns from the editor and reviewers. However, the paper writing quality is still far from satisfactory. Some examples are listed below, though I cannot list them all. I would suggest the authors asking some fluent writers to proofread the entire paper, otherwise, it would be difficult to give the final "Accept" decision.

Sec 5
- line 498: "is" proven
- line 499: "and" combine "it with" other... "models"

---

## Round 0.4 · Minor Revisions

The paper has improved the writing and thanks for the efforts.

However, as suggested by the Section Editor, the authors should better address concerns from the previous reviews. I looked at the reviews again and agree with this. Specifically, for "-" results in the tables, the authors should try to fill them and if the re-implementation is not feasible, there should be also clear explanations.

---

## Round 0.5 · Minor Revisions

It appears the previous comment is not addressed in this version. Since the submission is very close to publishing, I am reminding the authors to address them. Specifically, for empty ("-") results in the tables, the authors should try to fill them, or if the re-implementation is not feasible, there should be some explanations in the text. Please highlight the main changes in color in the next version. Thanks.

---

## Round 0.6 · accepted · Accept

Thanks for addressing all the concerns. The paper shows consistently better results with the proposed method compared to the prior work, which establishes a good benchmark in the community.